# COVID-19 Incidence Proportion as a Function of Regional Testing Strategy, Vaccination Coverage, and Vaccine Type

**DOI:** 10.3390/v15112181

**Published:** 2023-10-30

**Authors:** Areg A. Totolian, Viacheslav S. Smirnov, Alexei A. Krasnov, Edward S. Ramsay, Vladimir G. Dedkov, Anna Y. Popova

**Affiliations:** 1Saint Petersburg Pasteur Institute, 197101 St. Petersburg, Russia; totolian@spbraaci.ru (A.A.T.); vssmi@mail.ru (V.S.S.); dr.krasnov_28@mail.ru (A.A.K.); warmsunnyday@mail.ru (E.S.R.); 2Federal Service for Supervision of Consumer Rights Protection and Human Welfare, 127994 Moscow, Russia; depart@gsen.ru

**Keywords:** COVID-19, SARS-CoV-2 vaccines, SARS-CoV-2 testing coverage, COVID-19 incidence proportion, SARS-CoV-2 vaccination coverage, SARS-CoV-2 herd immunity

## Abstract

**Introduction:** The COVID-19 pandemic has become a serious challenge for humanity almost everywhere globally. Despite active vaccination around the world, the incidence proportion in different countries varies significantly as of May 2022. The reason may be a combination of demographic, immunological, and epidemiological factors. The purpose of this study was to analyze possible relationships between COVID-19 incidence proportion in the population and the types of SARS-CoV-2 vaccines used in different countries globally, taking into account demographic and epidemiological factors. **Materials and methods:** An initial database was created of demographic and immunoepidemiological information about the COVID-19 situation in 104 countries collected from published official sources and repository data. The baseline included, for each country, population size and density; SARS-CoV-2 testing coverage; vaccination coverage; incidence proportion; and a list of vaccines that were used, including their relative share among all vaccinations. Subsequently, the initial data set was stratified by population and vaccination coverage. The final data set was subjected to statistical processing both in general and taking into account population testing coverage. **Results:** After formation of the final data set (including 53 countries), it turned out that reported COVID-19 case numbers correlated most strongly with testing coverage and the proportions of vaccine types used, specifically, mRNA (V1); vector (V2); peptide/protein (V3); and whole-virion/inactivated (V4). Due to the fact that an inverse correlation was found between ‘reported COVID-19 case numbers’ with V2, V3, and V4, these three vaccine types were also combined into one analytic group, ‘non-mRNA group’ vaccines (Vnmg). When the relationship between vaccine type and incidence proportion was examined, minimum incidence proportion was noted at V1:Vnmg ratios (%:%) from 0:100 to 30:70. Maximum incidence proportion was seen with V1:Vnmg from 80:20 to 100:0. On the other hand, we have shown that the number of reported COVID-19 cases in different countries largely depends on testing coverage. To offset this factor, countries with low and extremely high levels of testing were excluded from the data set; it was then confirmed that the largest number of reported COVID-19 cases occurred in countries with a dominance of V1 vaccines. The fewest reported cases were seen in countries with a dominance of Vnmg vaccines. Conclusion: In this paper, we have shown for the first time that the level of reported COVID-19 incidence proportion depends not only on SARS-CoV-2 testing and vaccination coverage, which is quite logical, but probably also on the vaccine types used. With the same vaccination level and testing coverage, those countries that predominantly use vector and whole-virion vaccines feature incidence proportion that is significantly lower than countries that predominantly use mRNA vaccines.

## 1. Introduction

Despite large-scale control measures, both medical and non-medical, COVID-19 incidence proportion in the 3rd year of the pandemic remained high. Globally (as of 14 June 2023), there have been 767,984,989 confirmed cases, including 6,943,390 deaths, reported to the WHO. As of 12 June 2023, a total of 13,397,334,282 vaccine doses have been administered [1]. The daily number of confirmed COVID-19 cases (as of 13–14 May 2022, the data collection period for this study) was 32.53 cases per 1 million population in Russia. In other regions, the values (new cases per day per million) were 509.1 in France; 437 in Finland; 345.2 in Belgium; and 439.5 in the European Union (overall). On the other hand, 62.3 cases were noted in Lithuania, but only 19.4 in Turkey, and no new illnesses were detected in Kyrgyzstan or Tajikistan during this period [2]. Although the reference states “Due to limited testing, the number of confirmed cases is lower than the true number of infections,” there is nevertheless a significant gap in the daily number of confirmed cases in different countries. There are likely several reasons for such significant differences.

Assigning primary importance to population would be reasonable. It could be assumed that highly populated countries would feature a higher incidence proportion. In the pandemic, however, this factor was not so significant. For example, the population of Russia was 145,478,097 (as of 13 May 2022), while that of France was 67,390,000 [3]. In France, however, the daily increase in patients in mid-May was 15.6-fold higher than in Russia.

Population density could be another factor. It is believed that the greater number of contacts within a high-density region can become a factor contributing to more active pathogen spread [4]. However, this indicator turned out to be ambiguous. As an example, Lithuania and Latvia feature densities of 40.6/km^2^ and 28.7/km^2^, respectively. However, the daily increase in patients in Lithuania was 62.3 people per 1 million, and Latvia’s was 126.0 [2].

The third reason for viral spread can be population mobility. It has been repeatedly shown that the reason for the rapid spread of SARS-CoV-2 was the active global movement of tourists. Thus, severe restrictive measures were implemented early in the pandemic in many countries.

Vaccination has become important in limiting viral spread [5,6]. In parallel with the introduction of vaccines developed at unprecedented speeds, a fairly strong anti-vaccination movement formed. Significant time and effort were required to counter baseless messaging that put lives at further risk. PCR testing has also played a key role. Widespread use of diagnostic tests contributed to the timely detection of overt and asymptomatic infections, with the latter often prevailing [7,8].

SARS-CoV-2 belongs to a family of respiratory viruses characterized by high genetic variability. The progenitor Wuhan strain circulated globally in the first half of 2020. Since December 2020, it was replaced by variant B.1.1.7 (alpha) [9], followed by more pathogenic beta (501.V2/B.1.351) and gamma (B1.128) variants, which had the phenomenon of immunological escape [10,11]. In February 2020, there was a P.1 strain (501Y.V3), which spread rapidly despite the start of clinical trials of the first anti-coronavirus vaccines (mRNA platform) at that time [12]. This was a clear signal that new SARS-CoV-2 vaccines will not necessarily be effective against all mutant viral variants. In December 2020, the B.1.617.2 variant (Delta) was detected in India for the first time, displacing other genetic variants [13,14,15]. The highly pathogenic B.1.1.529 variant (Omicron) was the last to be detected [14,16]. It was detected in vaccinated individuals and turned out to be 2.8-fold more contagious than the Delta variant [14,17,18,19].

Vaccination coverage is another factor. In addition to post-infectious immunity, collective immunity is influenced by the vaccination coverage of the population (formation of post-vaccination immunity). Specific features of vaccines used in different countries could be a significant factor that determines local incidence proportion. Existing vaccines can be divided into four types by platform: mRNA vaccines; vector vaccines; whole-virion vaccines; and peptide vaccines. The first three are the most common globally.

The first vaccine, mRNA-1273, was developed by Moderna in early spring 2020; its clinical trials began in China on 16 March 2020 [20]. Pfizer/BioNTech presented BNT162b2 based on a similar platform [21,22]. This family of preparations has been shown to promote the production of a limited set of anti-spike antibodies (Abs) and cellular responses, with active responses of CD4^+^ T cells in particular [23]. An mRNA vaccine forms a stable immune response by the 7th day after the first injection, reaching a maximum on the 14th day after the second injection. The response most often persists for about three months [24]. This immunity can be overcome by a genetically different viral variant, such as B.1.1.529 (Omicron) [17]. This is exactly what happened in Israel. Even with almost total double-immunization with BNT162b2, a significant increase in COVID-19 was noted during the outbreak caused by B.1.1.529; additional booster vaccinations were required [25].

Vector vaccines are another group consisting of a vector with an embedded fragment of the viral genome. Upon entering a cell, they generate an immunogen (spike protein) that induces an immune response, including Ab production and cellular immunity, polarized towards Th1 IFNγ [26]. Vectors used have included two human adenoviruses, Ad26 and Ad5, in the Gam-COVID-Vac vaccine and chimpanzee adenovirus ChAdOx1 (ChAdOx1 nCoV-19 vaccine, AZD1222). In many countries, the non-replicating vector vaccines Convidicea (CanSino Biologics) or Janssen COVID-19 Vaccine (Johnson & Johnson) were actively used [27,28,29,30]. Comparative studies indicate similar safety and efficacy of mRNA and vector vaccines [31]. However, the Ab spectrum and their duration of circulation is longer in vector vaccines than in mRNA vaccines [32].

Inactivated whole-virion vaccines represent a traditional development technology used in many countries. They have a good safety profile and a strong Ab response, but their relatively low immunogenicity requires the use of an adjuvant and multiple vaccinations [33]. Among them, the Chinese vaccines Sinovac/CoronaVac, Sinopharm BBIBP-CorB, and Covaxim are the most common [34,35,36,37]. A whole-virion inactivated vaccine, CoviVac, has been registered in Russia [38]. Similar vaccines have been developed and registered in other countries. Some examples include: QazCovid-in^®^ in Kazakhstan [39]; BBV152/COVAXIN in India [37]; Soberana 02 and Abdala in Cuba [40]; and BIV1-CovIran in Iran [41]. It has been shown that all of these are high-quality vaccines capable of generating full-fledged immune protection and forming the widest spectrum of immune response [33,38,42,43]. The purpose of this study was to analyze possible relationships between COVID-19 incidence proportion and vaccine types used in different countries, taking into account demographic and epidemiological factors.

## 2. Materials and Methods

### 2.1. Formation of the Initial Database

For this analysis, we have created a database in which was collected demographic and immunoepidemiological information about the COVID-19 situation in 104 countries as published in official sources and repositories. Information on incidence proportion, testing coverage, vaccination coverage, dates of the vaccine company beginning, and vaccine ratios used (as of 10–20 May 2022) was taken from Ritchie et al. (https://ourworldindata.org, accessed on 13 May 2022) [2]; Official Coronavirus Statistics (https://gogov.ru/articles/covid-v-stats, accessed on 29 October 2022) [44]; Coronavirus Monitor (https://coronavirus-monitor.info, accessed on 10–20 May 2022) [1]; the World Health Organization (https://covid19.who.int, accessed on 10–20 May 2022) [45]; and the Pan American Health Organization (https://www.paho.org, accessed on 15–20 May 2022) [46].

When necessary, we used official information from the government websites of a number of states. We also used data on population size and density in different countries provided by the United Nations Department of Economic and Social Affairs (UNDESA) [47]. The collected information was brought together in a single database, shown in Appendix A. As no contact with the national populations listed in Appendix A was expected, and the analysis used previously published information in the aforementioned official sources, ethics committee approval was not required.

### 2.2. Study Design

The main criteria for inclusion in the initial database were availability of information about the vaccines used and data on the relative usage of each among all vaccinated. The study’s design is schematically presented in Figure 1. The baseline included the following information for each country/region: population size and density; SARS-CoV-2 testing coverage (tests per 1000 population); incidence proportion (cumulative number of registered cases per 1 million population) after the beginning of national vaccination company; vaccination coverage (%); and a list of vaccines used, including the share of each among all vaccinated (%).

Since we initially collected data on all countries for which information was available (vaccines, relative share), further stratification of the initial data set (by population size, vaccination coverage) was necessary. The final data set was subjected to statistical processing, both in general and taking into account testing coverage of the population.

### 2.3. Statistical Analysis

Statistical analyses were performed, depending on the nature of the data distribution, using parametric and non-parametric methods in the Excel (2010) and Statistica (ver.12, Windows) applications. The results were presented as median (Me) and interval [Q25–Q75]. Rank correlation analysis was performed using the Spearmen method. Multivariate, discriminant, and regression analyses were performed using Statistica (ver.12, Windows). Calculation of statistically significant differences was carried out according to the Mann–Whitney test. When necessary, Bonferroni correction for multiple comparisons was applied. Differences were designated as significant, unless otherwise indicated, at the *p* < 0.05 level.

## 3. Results

### 3.1. Characteristics of the Original Data Set and Correction

Figure 2 shows the information collected for 104 countries, including population size and density; testing coverage (tests per 1000 population); incidence proportion (cumulative number of registered cases per 1 million population) after the beginning of national vaccination company; vaccination coverage (%); and enumeration of vaccines used, including the share of each among all vaccinated (%). The collected primary information is summarized in Appendix A.

#### 3.1.1. National Populations

This indicator varied widely (Appendix A), with maximum values in China (pop. 1,439,324,000) and India (pop. 1,380,004,000), alongside minimum values in Montserrat (pop. 5000) and the Falkland Islands (pop. 3000). Globally, there are more than 200 countries, including groups of large and small nations. United Nations nomenclature [48] designates these as largest (>100 million), large (40–100 million), medium (20–40 million), small (1.5–20 million), and smallest (0.5–1.5 million). Finally, dwarf microstates (<0.5 million) are distinguished in a number of cases [49].

According to this classification, 104 countries were divided into the relevant groups (Table 1). According to correlation analysis, population size did not correlate with COVID-19 incidence proportion in any way (r= −0.144; *p* > 0.1). For this reason, base adjustment for this indicator was minimal, consisting only of the removal of ‘dwarf states’. Hence, for initial adjustment, we excluded countries with a population of 500,000 or less from the study’s data sample. In result, 86 countries remained in the database.

#### 3.1.2. Vaccination Coverage

The next indicator, of fundamental importance for the validity of analysis, was vaccination coverage. The number vaccinated relative to overall population (%) was used as an indicator reflecting vaccination level. In the initial data set, coverage fluctuated over a wide range (Appendix A): from 0.1% (Burundi) to 96.9% (Chile). Therefore, the sample was stratified according to this indicator. The lower threshold for stratification was 50% vaccination coverage. The legitimacy of this approach is primarily due to epidemiological factors. It is known that vaccination coverage thresholds largely determine COVID-19 epidemic spread dynamics [4]. Depending on prevailing conditions, this indicator can vary from 30% to 85% [50,51].

Here, we used 50% as the minimum effective threshold for vaccination coverage. This level was also substantiated by a correlation analysis between incidence proportion and vaccination coverage in the original data set. Significant correlation was found (r = 0.5112) with strong significance (*p* < 0.00001). Subsequent regression analysis showed that this dependence is described by a logarithmic curve equation of the form y = 8.1469ln(x) − 31.999 (Figure 3A). The resulting regression curve consists of two branches. The left vertical branch describes the initial phase of immune response formation; it is associated with incidence proportion level with a probability of *p* = 0.00001. The choice of this threshold is due to the transition point of the regression curve’s ascending branch to a flat section, established in the process of regression analysis.

In order to exclude the impact of this phase on incidence proportion, a correction was made to exclude all countries with less than 50% vaccination coverage from the data set. Thus, the second base correction was exclusion of such countries (coverage < 50%) from the set. As a result, 33 countries were excluded, leaving 53 countries in the final data set (Appendix A). As such, the range of vaccination levels was significantly narrower, with a maximum in Chile (96.9%) and a minimum in Venezuela (50.2%). Following correction, the logarithmic regression transformed into a linear form, described by the equation y = −9E − 06x + 76.044 (Figure 3B). At the same time, the coefficient of determination value (R²) decreased from 0.4044 (Figure 3A) to 0.0002 (Figure 3B). This indicates an absence of any significant relationship between incidence proportion and vaccination coverage (r = −0.1157; *p* > 0.1).

#### 3.1.3. Population Density

In the initial data set, density also showed high heterogeneity (Appendix A), with a maximum in Monaco (26,152.3/km^2^) and a minimum in Greenland (0.1/km^2^). In the final data set, the population density range was significantly narrower (Appendix A), with a maximum in China Hong Kong (7082.1/km^2^) and a minimum in Australia (3.3/km^2^). Density also did not correlate with COVID-19 incidence proportion in any way (r = 0.09; *p* > 0.1). For this indicator, the base was not adjusted.

#### 3.1.4. COVID-19 Incidence Proportion

The cumulative number of reported cases per 1 million population was used as an integral indicator of incidence proportion. In the initial data set, this indicator varied over a wide range (Appendix A), from 664,914 (Faroe Islands) to 237 (Cook Islands). In the final data set, the incidence proportion range was significantly narrower (Appendix A), with a maximum in Cyprus (522,916) and a minimum in China (536).

#### 3.1.5. SARS-CoV-2 Testing Coverage

Testing coverage was assessed as the cumulative number of tests per 1000 population. In the initial data set, this information was not available for 25 countries. For the remaining 79 countries, this indicator varied widely (Figure 4), from 32,860 (Cyprus) to 10 (Niger). In the final data set, information on this indicator was missing for only three countries, and the minimum value increased to 52 (Sierra Leone). Testing coverage was strongly correlated with COVID-19 incidence proportion. In the initial data set, the relationship was strong (r = 0.5949; *p* < 0.000001). The relationship was also confirmed in the final set (r = 0.580; *p* < 0.001). For this indicator, the base was not adjusted. However, the presence of a stable, highly significant correlation with incidence proportion justified the need for further analysis taking it into account.

#### 3.1.6. Vaccines Used in Different Countries

During preliminary work, it was found that different vaccines (created on four main platforms) were used in different countries. In some, vaccines from the mRNA platform were fully or partially used. In other countries, various platforms (vector, protein/peptide, whole-virion) were used. In most countries, several vaccines created on different platforms were used. Unfortunately, there was no detailed quantitative information on the vaccines used for many countries, which significantly complicated analysis. All vaccines used were divided into four groups.

The first group included preparations based on the mRNA platform (mRNA-1273, BNT162b2). Among them, the Pfizer vaccine accounted for 80.3%, and the Moderna vaccine accounted for 19.7%. This group was abbreviated as V1.

In the 2nd group, all vector vaccines were combined: Johnson & Johnson; ChAdOx1 (Oxford-AstraZeneca, Covishield); Gam-COVID-Vac (Sputnik V); and Convidecia (CanSino Biologics). In the final data set, the Oxford/AstraZeneca vaccine accounted for the largest share (75.6%). Gam-COVID-Vac (Sputnik V) represented 12.3%, and the Johnson & Johnson vaccine represented 10.7%. This group was abbreviated as V2.

Peptide and protein vaccines were combined into group V3 (Novavax, EpiVacCorona, Abdala, Corbevax); their incidence proportion was less than 1%. Whole-virion vaccines were grouped as V4 (Sinovac, Sinopharm, Soberana, COVIran Barekat, Covaxin, FAKHRAVAC, QazVac, IMBCAMS, KCONVAC, CoviVac). In terms of the number of vaccines developed, V4 is the largest group. In it, Sinopharm stood out (54.3%) alongside Sinovac (40.6%) and Covaxin (4%).

When information was available on each of the four vaccine types for a country, their share of total vaccines was calculated. In terms of the incidence proportion of vaccine types (platforms used), the leaders were V1 (59.08%), V2 (27.75%), and V4 (12.89%). Type V3 accounted for only 0.28%.

### 3.2. Characteristics of the Final Data Set

The final data set included 53 countries, the distribution of which (by group) is presented in Table 2. General characteristics of the indicators used (population size and density, SARS-CoV-2 testing coverage, reported COVID-19 case numbers, vaccination coverage, enumeration of vaccines used) have been presented above (Section 3.1). Correction made it possible to establish the final structure of correlations in the set (Table 2). Multiple correlation analysis revealed a number of strong statistical relationships between ‘reported COVID-19 case numbers’ and other sample parameters (Table 2).

In the pair ‘reported COVID-19 case numbers’–‘population size’, a weak inverse correlation was found (r = −0.282; *p* < 0.05). Correlations were not found in two other pairs: ‘reported COVID-19 case numbers’–‘population density’; and ‘reported COVID-19 case numbers’–‘vaccination coverage’. As mentioned earlier, a strong correlation was found in the pair ‘reported COVID-19 case numbers’–‘testing coverage’ (r = 0.580; *p* < 0.001). Therefore, further analysis was carried out taking into account the number of tests per 1000 population.

In addition to those already listed, correlations were established between ‘reported COVID-19 case numbers’ and the share of vaccine types (V1, V2, V3, V4) used. The strongest, significantly positive, relationship between was found for V1 (r = 0.712; *p* < 0.0001). An inverse correlation was found for other vaccine types (significant for V2, V4), and for V3 this relationship was in the nature of a trend. Given the unidirectional correlation coefficients between ‘reported COVID-19 case numbers’ with V2, V3, and V4, we considered it reasonable to combine these three vaccine types into one group, designated ‘non-mRNA group vaccines’ (Vnmg) for further analysis, as shown in Appendix A.

### 3.3. Influence of Share V1 and Vnmg on ‘Reported COVID-19 Case Numbers’ in Different Countries Globally

As already shown, group V1 and Vnmg vaccines featured opposite relationships with ‘reported COVID-19 case numbers’ (Table 3). In different countries, their ratios varied from 0 to 100% (Appendix A). In this regard, we hypothesized that a certain factor influencing ‘reported COVID-19 case numbers’ could be the ratio of vaccine types used in different countries. To test this assumption, a continuous axis of V1:Vnmg ratios was plotted from 0 to 100% (Figure 5). Analysis of these ratios (%:%) permitted distribution of the sample to countries in which V1 vaccination prevailed (80:20, 90:10, 100:0); countries where V1 and Vnmg usage was nearly equal (60:40, 50:50, 40:60); and countries in which V2 vaccination predominated (0:100, 30:70). The lowest ‘reported cumulative COVID-19 case numbers’ were found at V1:Vnmg ratios from 0:100 to 30:70; the highest cumulative case numbers were found at ratios from 80:20 to 100:0.

Based on the results, three subgroups of countries were formed. The 1st subgroup (V1 > Vnmg) included countries with V1:Vnmg ratios from 70:30 to 100:0 (*n* = 30). The 2nd subgroup (V1 ≈ Vnmg) contained the V1:Vnmg ratios from 40:60 to 60:40 and included six countries. The 3rd subgroup (V1 < Vnmg) included V1:Vnmg ratios from 0:100 to 30:70 and included 17 countries (Figure 6).

The highest ‘reported COVID-19 case numbers’ (hereafter per million, unless noted) were noted in the 1st subgroup where V1 > Vnmg (316,427 [199,143–408,993]). The lowest level was in the 3rd subgroup with V1 < Vnmg (36,980 [7464–95,956]). Moreover, this indicator in the V1 < Vnmg subgroup was significantly lower than in the V1 > Vnmg subgroup (*p* = 0.000001) and the V1 ≈ Vnmg subgroup (*p* = 0.0099). The V1 ≈ Vnmg subgroup had an intermediate level of ‘reported COVID-19 case numbers’ (160,698 [78,556–270,386]). Statistical analysis of the significance of differences (between the compared groups according to three statistical criteria) is given in Appendix A.

At the same time, there were no statistically significant differences in ‘reported COVID-19 case numbers’ between subgroups V1 > Vnmg and V1 ≈ Vnmg (*p* = 0.0358). Taking into account the multiplicity of intergroup comparisons, Bonferroni’s correction for multiple comparisons was applied to determine the critical *p* value. Given that each group was compared twice, the critical *p* value after applying the correction was *p* ≤ 0.025. The *p* values in both comparison cases were less than the critical value obtained using the Bonferroni correction. Thus, reported COVID-19 case numbers in the V1 < Vnmg subgroup (adjusted for multiple comparisons) were significantly lower than in the other two groups. The data obtained confirmed the initial assertion that V1 and Vnmg were in opposite relationships with ‘reported COVID-19 case numbers’ in the studied populations.

### 3.4. Dependence of ‘Reported COVID-19 Case Numbers’ on Vaccine Type, Taking into Account SARS-CoV-2 Testing Adherence

Reported COVID-19 case numbers in different countries can largely depend on testing coverage. This is particularly evidenced by a strong correlation between these indicators (r = 0.58; *p* < 0.001) with a high level of determination, R^2^ = 0.677 (Table 3). To determine the legitimacy of such a dependence, we analyzed the exponential curve shown in Figure 7. Based on characteristics of the trend line, which reflects the dependence of incidence proportion on testing coverage, the study sample was empirically divided into three groups. Grouping validity was checked using discriminant analysis, which included two independent variables: ‘testing coverage’ and ‘reported COVID-19 case numbers’. As a result of a stepwise change in the structure of the groups, it was possible to formulate a discriminant function with a high degree of reliability for both independent variables: for the variable ‘testing coverage’ (*p* < 0.000001); and for the variable ‘incidence proportion’ (*p* = 0.000014).

When assessing classifying ability, this discriminant model made the least number of classification errors: 7 errors out of 50 classification acts. Discriminant analysis data are presented in Appendix A. As a result of discriminant analysis, three groups were identified with the following testing coverage ranges (tests per 1000 people): group 1 with ≤972 (*n* = 13); group 2 with from 1007 to 7348 (*n* = 33); and group 3 with ≥9380 (*n* = 4). Although the number of tests is significant, it is not, however, the only indicator that influences ‘reported COVID-19 case numbers’. Another factor may be the type of vaccines used. In each group, the effect of V1:Vnmg vaccine ratio on ‘reported COVID-19 case numbers’ was assessed according to the subgrouping described earlier (Section 3.1).

In the first group, which included countries with a low level of testing, division into V1 > Vnmg and V1 < Vnmg did not reveal statistically significant differences, which seems quite logical for countries where there is less than one test per resident. The third group (countries with an extremely high level of testing) was excluded from further analysis due to its small size.

The second group included countries with testing levels ranging from 1007 to 7348 tests per 1000 people (Figure 8A). This group represented the main sample, within which the correlation coefficient between ‘reported COVID-19 case numbers’ and the level of testing was r = 0.148 (*p* > 0.1); this proves that there is no relationship between these two indicators.

In this main sample group, the ‘reported COVID-19 case numbers’ indicator (Figure 8B) was 302,160 (219,520–302,867) for countries in subgroup 1 and 103,976 (84,633–157,635) for countries in subgroup 3, with a high level of significance (*p* = 0.000244) for the difference between subgroups 1 and 3 (Appendix A).

For subgroup 2 countries with V1 ≈ Vnmg, this indicator occupied a middle position, amounting to 161,440 (159,957–265,490). Differences between subgroups 1 and 2 were significant (*p* = 0.027391); they were absent between subgroups 2 and 3 (*p* = 0.21). Thus, when the important variable ‘testing coverage’ was excluded from the data set, higher ‘reported COVID-19 case numbers’ are noted among countries with predominant use of V1 vaccines. Among the populations of countries where Vnmg vaccines were more widely used, ‘reported COVID-19 case numbers’ were lower by an average of 2.8-fold.

Since Russia was in the 2nd group, where ‘reported COVID-19 case numbers’ were several times lower than in neighboring European countries, we separately analyzed countries comparable to Russia in terms of testing coverage. To this end, a sample of countries was formed such that Russia’s testing coverage occupied the median value (2001 tests per 1000 people); the interquartile interval was 1697–2602. This interval included 13 countries (Figure 9A). Similar to analysis of the entire sample, significant differences (*p* = 0.041259) were found between subgroup 1 (317,507 [225,147–399,973]) and subgroup 2 (130,806 [117,391–144,220]) (Figure 9B). Moreover, it should be noted that, in Russia, ‘reported COVID-19 case numbers’ were 1.6-fold lower than in Finland, which features the lowest level of this indicator among subgroup 1 countries.

During formation of the data set, two aspects were not analyzed: duration of the post-vaccination period and specifics of viral genetic variants circulating in the population (with varying ability to evade immunity). We can only note the observed decrease in ‘reported COVID-19 case numbers’ in countries where vector and whole-virion vaccines were used. Possible reasons for this phenomenon should probably be the subject of further research.

## 4. Discussion

The COVID-19 pandemic has become one of the biggest challenges of the 21st century. Suddenly appearing on 31 December 2019 in Wuhan (PRC), it covered most of the world within a matter of weeks. The lack of any effective antiviral drugs and specific vaccines prompted most countries globally to respond in the form of unprecedented restrictive measures. On the other hand, the lack of effective therapeutic and prophylactic agents contributed to the unprecedented activity of the scientific community. This resulted in the rapid development and implementation of a whole series of vaccines on various platforms, with no parallels in the history of medicine [33,53,54]. Four of them are the most widespread: messenger RNA-based vaccines (designated as group V1 in this work) [55,56]; vector vaccines (V2) [57,58]; peptide and protein vaccines (V3); and inactivated, whole-virion vaccines (V4) [41,42,43].

It should be noted that most vaccines were created on the basis of the progenitor Wuhan strain [59]. Meanwhile, already in 2020, new genetic variants of the virus began to appear, characterized by more pathogenic properties. As a result, data appeared on the ability of new viral variants to overcome the adaptive immunity created by vaccines [60,61,62]. In addition, it has been shown that various vaccines form post-vaccination immunity differently; they can differ both in duration and in conferred resistance to new viral genetic variants [63,64]. All of the above may have contributed to the varying COVID-19 incidence proportion around the world.

In this regard, it seemed relevant to assess the relationship between COVID-19 incidence proportion in the population and SARS-CoV-2 vaccine types used in different countries globally, taking into account demographic (population size, density) and immunobiological (testing and vaccination coverage) factors. The work was sequentially performed, according to algorithm, in several stages (Figure 1). The main results are presented in Figure 2. To this end, the necessary information was collected from available sources for 104 countries. The main inclusion criterion was availability of information on the vaccines used, including their proportional contributions to overall vaccination.

The collected information was subsequently summarized in a general table (Appendix A), and initial analysis showed high heterogeneity in all indicators. Heterogeneity in terms of demographic indicators was seen by the presence of all country groups in the initial data set (according to UN classification); only dwarf states were subsequently excluded. For the goals set in the research, vaccination coverage was a critical indicator, initially ranging from 0.1% in Burundi to 96.6% in Chile and South Africa (Appendix A). For valid analysis, countries with vaccination coverage below 50% were further excluded. The People’s Republic of China (PRC) was included in the data set despite local peculiarities in controlling the COVID-19 epidemic, specifically the PRC’s overall ‘zero COVID-19 strategy’ in the event of outbreaks. This strategy is essentially unique and is not applied in other countries of the world.

Following preparation of a final data set of 53 countries, reported COVID-19 case numbers were found to be most strongly correlated with testing coverage and proportion of vaccine types used. Due to the fact that an inverse correlation was found between ‘reported COVID-19 case numbers’ with V2, V3, and V4, we considered it possible to combine these three types of vaccines into one group (non-mRNA group vaccines, Vnmg) for further analysis. Moreover, to offset the impact of SARS-CoV-2 testing on reported COVID-19 case numbers, countries with low testing rates (<1 test/person) and countries with extremely high testing rates were excluded. The resulting analyzed sample represented 33 countries. When analyzing reported COVID-19 case numbers within this sample, it was shown that the highest numbers occur in countries with V1 vaccine (mRNA) dominance. The lowest was seen in countries with Vnmg (vector, peptide/protein, whole-virion/inactivated) vaccine dominance (Figure 8).

For the most appropriate comparison of Russia with other countries, a data set was formed with a specific median level of ‘testing coverage’ (2001 per 1000 people). Analysis showed that, like analysis of the overall sample, there were significant differences between countries with V1 vaccine (mRNA) dominance and countries with Vnmg dominance.

Moreover, it should be noted that, despite exactly the same testing levels, ‘reported COVID-19 case numbers’ for Russia (with Vnmg usage exclusively) were 1.6-fold lower than, for example, in Finland. Russia’s low numbers are remarkable given the fact that Finland itself already represents with the lowest case numbers among 10 countries with the dominant use of mRNA vaccines (Figure 9). The results show that, in countries where non-mRNA vaccines have been used, ‘reported COVID-19 case numbers’ are significantly lower than in countries with dominant use of mRNA vaccines.

A chart summarizing information for all 53 countries regarding reported COVID-19 case numbers, and their dependence on vaccine types used, is presented in Figure 10. In addition to polar groups (nations dominated by V1 or Vnmg), the group of countries with equal proportions of mRNA and non-mRNA vaccines is of particular interest. As expected, reported COVID-19 case numbers for these countries are middle values.

The South Asian country of Bhutan stands apart because unique experience was gained through combined use of different vaccine types. The first immunization was carried out with an mRNA vaccine; the second immunization used a vector vaccine [65]. At the same time, high vaccination coverage and the lowest ‘reported COVID-19 case numbers’ were achieved among countries in this group. However, case numbers in Bhutan were higher than in the neighboring regional countries of India and Nepal. To be fair, in the latter two countries, testing coverage was less than one test per person.

A similar situation developed in East Asian countries. The maximum incidence proportion was registered in South Korea, which used mainly mRNA vaccines. The minimum was registered in China with the dominant use of whole-virion vaccines. China Hong Kong, with an equal proportion of mRNA and non-mRNA vaccines, featured an intermediate incidence proportion. This is despite the fact that the minimum testing coverage was in South Korea, and in China and China Hong Kong the level of testing was equally high. Interestingly, in Japan, the incidence proportion was the lowest among countries in the region; this was combined with a very low level of testing for highly developed countries. On the other hand, in Israel, where testing coverage was even slightly lower than in China Hong Kong and China, ‘reported COVID-19 case numbers’ were 2.8-fold higher than in China Hong Kong and three orders of magnitude higher than in China.

In a number of European countries in which mRNA vaccines dominated (France, Portugal, Luxembourg, Estonia, Slovenia, Switzerland), incidence proportion was 18–45% higher than in the UK. The UK had an even vaccine-type ratio (mRNA vs. non-mRNA), and testing coverage was equal to, or higher (1.5- to 3-fold) than, the listed countries.

In another group of four European countries with the same testing level, maximum incidence proportion was registered in Germany and Croatia, which mainly used mRNA vaccines. The minimum was in Belarus with dominant use of vector and whole-virion vaccines (2.6- to 3-fold less than Germany and Croatia). An intermediate level was seen in Hungary with an equal ratio of mRNA and non-mRNA vaccines (1.4- to 1.6-fold less than in Germany and Croatia).

The presented statistical calculations inevitably raise the question of what is the underlying cause of the difference noted. Comparative studies have convincingly shown the high efficacy of all currently approved vaccines [61,66]. However, it is impossible not to notice some differences that can affect incidence proportion. Regarding mRNA vaccines, one can agree with the opinion of a number of researchers who have shown that they are able to form rapid immunity in the early stages after immunization, persisting for 3 months, following which the use of a booster dose may be required [21]. As for vector vaccines (assigned to the Vnmg group), full-fledged immunity is formed by the 14th day after the second dose, but it exists for at least 6 months [28,30,67,68]. Regarding whole-virion vaccines, it is worth noting their lowest immunogenicity alongside their longest elicited immunity [69,70], which is closest to a post-infectious response.

The key parameter is probably the formation of stable herd immunity. Unfortunately, there are currently no published studies that have been conducted according to a single methodology with different countries globally when examining epidemic process dynamics as was performed here. In on our experience, assessment of SARS-CoV-2 collective immunity, carried out according to a single methodology [71] at different stages of the epidemic in Russia [8,72], Belarus [73], and Kyrgyzstan [74], showed that there is successful formation of herd immunity in those countries. Those countries were discussed in our earlier work, and usage of vector and whole-virion vaccines usage dominates in them.

On the other hand, in the work of Morens et al. [75], it was suggested that it is impossible to achieve long-term herd immunity with COVID-19 and therefore regular booster immunization is necessary. This is probably true primarily for mRNA vaccines, whose distribution currently dominates globally. In our opinion, the main reasons for this are breadth (spectrum) of the immune response and constant viral variability, through which new genetic variants appear regularly [76,77].

Vector vaccines, and even more so whole-virion vaccines, induce a significantly wider range of post-vaccination antibodies than mRNA vaccines [33,78]. Minimal diversity and maximal specificity were features embraced in the initial ideology of mRNA vaccine development. Undoubtedly, it is a very progressive technology that makes it possible to induce a narrow spectrum of post-vaccination antibodies, resulting in a decrease in the share of post-vaccination adverse reactions, including those of an autoimmune nature [79]. In conditions of high viral variability, however, the technology likely has a number of limitations, and the formation of a narrow Ab spectrum is more of a disadvantage than an advantage. As a result, the level of post-vaccination immunity persists for a short time, and its restoration is impossible without the introduction of a second vaccine dose.

In contrast, an immune response is formed to a much wider range of antigens and their epitopes with the use of vector and whole-virion vaccines. As a result, when post-vaccination immunity is attenuated, a repeat encounter with the virus, even a new genetic variant, leads to activation of the secondary immune response. As such, regular booster immunization is not required. A similar situation is observed with healthcare workers. Often, they have been ill only once, but regular restoration of post-infectious immunity ensues through periodic contact with infected carriers.

Thus, we have shown for the first time that ‘reported COVID-19 case numbers’ (per million population) apparently depend not only on SARS-CoV-2 testing coverage and vaccination coverage, which is quite logical, but also probably on the vaccine types used. With the same level of vaccination and testing coverage, countries using predominantly vector and whole-virion vaccines experienced significantly lower incidence proportion than countries predominantly using mRNA vaccines.

## 5. Limitations of the Study

When analyzing the assembled country database, we did not have data on the timing of primary or booster vaccinations. Therefore, we could not in any way assess post-vaccination immunity level at the time of information collection. Another factor is the uneven spread across countries of new genetic variants of the virus, which may feature different abilities to evade the immune response. Additionally, it should be noted that our study may be influenced by the differences in national approaches to the registration of COVID-19 incidence as well as the transportability of the data provided. Answering these and other questions would likely permit more accurate determination of the nature of post-vaccination morbidity. Perhaps this will be the subject of a separate study someday.

## Figures and Tables

**Figure 1 viruses-15-02181-f001:**
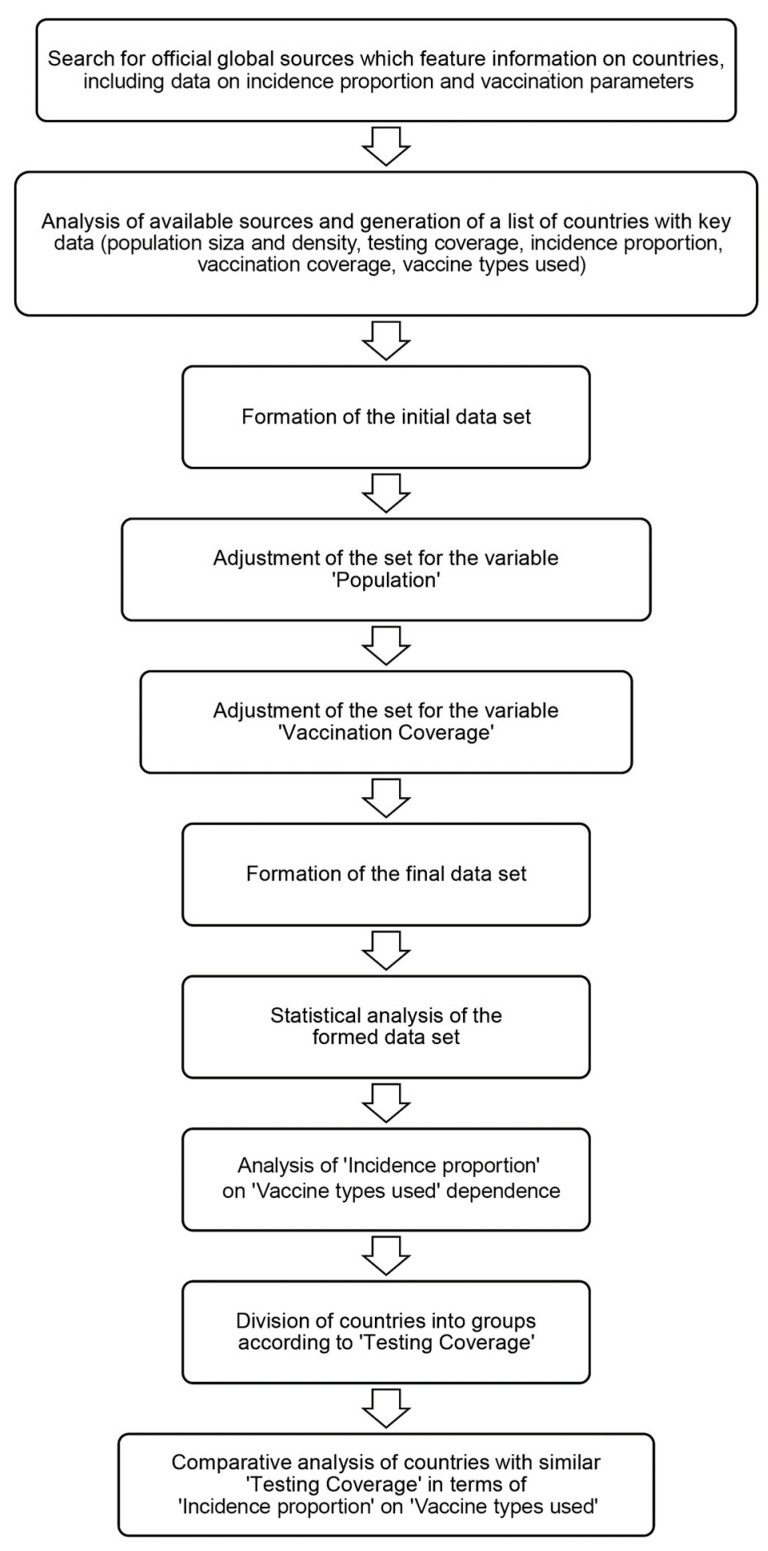
Research design workflow.

**Figure 2 viruses-15-02181-f002:**
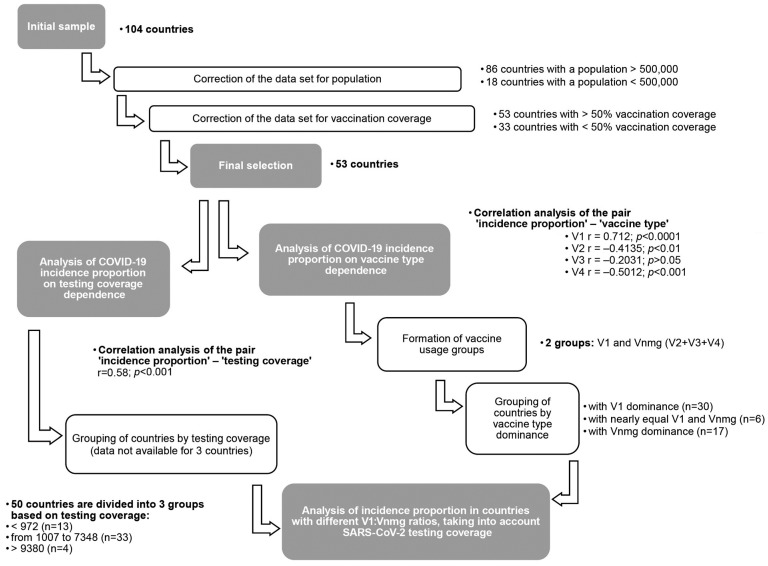
Study profile.

**Figure 3 viruses-15-02181-f003:**
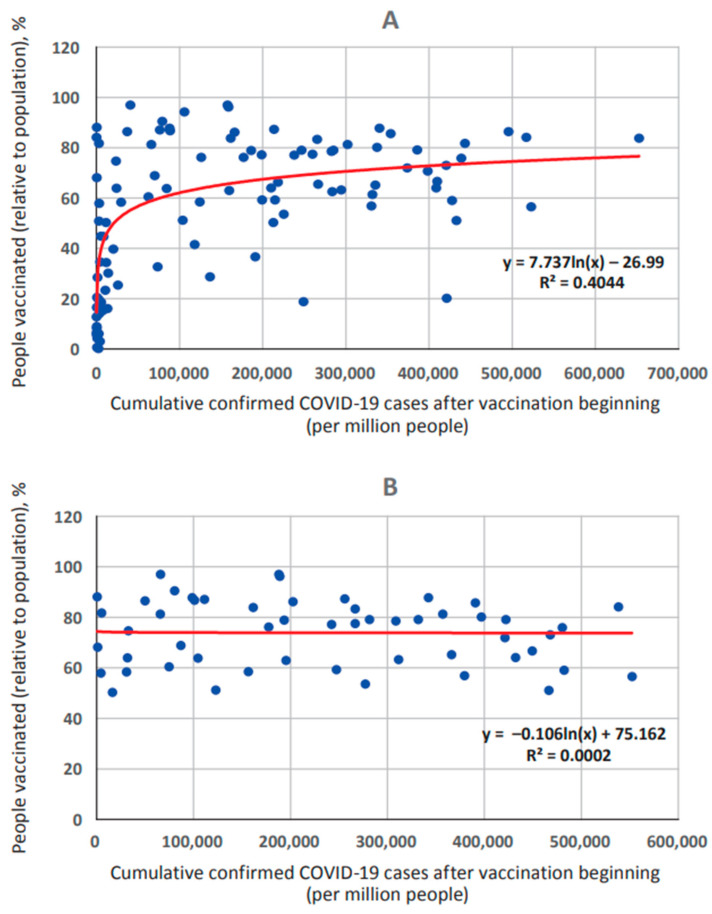
Relationship between reported COVID-19 case numbers and vaccination coverage. Key: (**A**)—initial data set (*n* = 104); (**B**)—final data set (*n* = 53). The regression equations are highlighted in red in the lower right.

**Figure 4 viruses-15-02181-f004:**
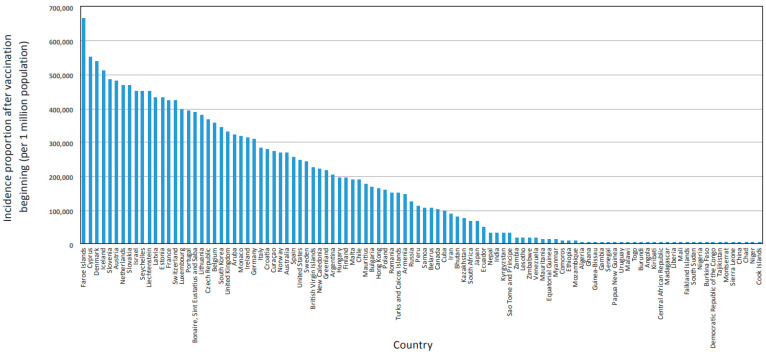
Distribution of countries (*n* = 79) included in the original data set (Appendix A) by testing coverage (cumulative number of tests per 1000 people). Note: No data available for 25 out of 104 countries. Source: Our World in Data [52]).

**Figure 5 viruses-15-02181-f005:**
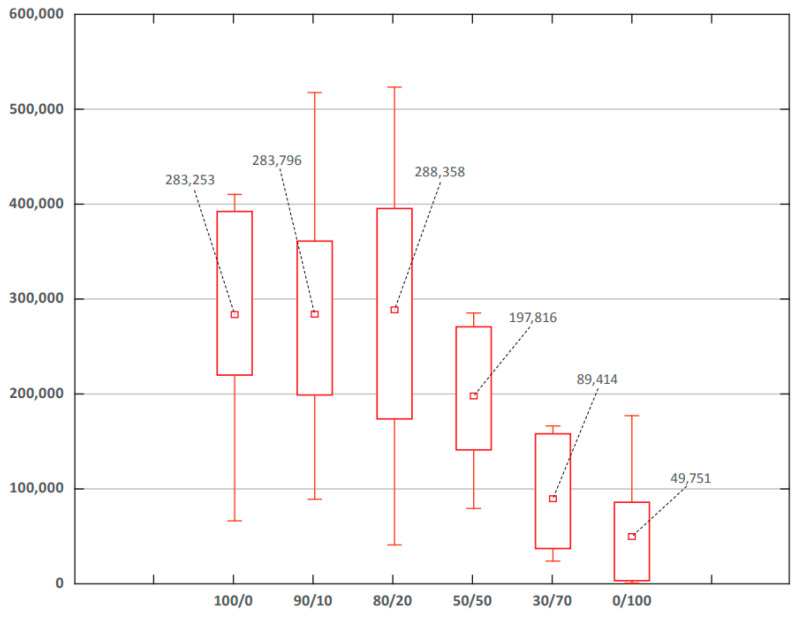
Reported COVID-19 cumulative case numbers in groups with different vaccine usage ratios (V1:Vnmg). Note: x-axis—V1:Vnmg ratio (%:%); y-axis—cumulative cases per 1 million population; numbers in boxes—median values.

**Figure 6 viruses-15-02181-f006:**
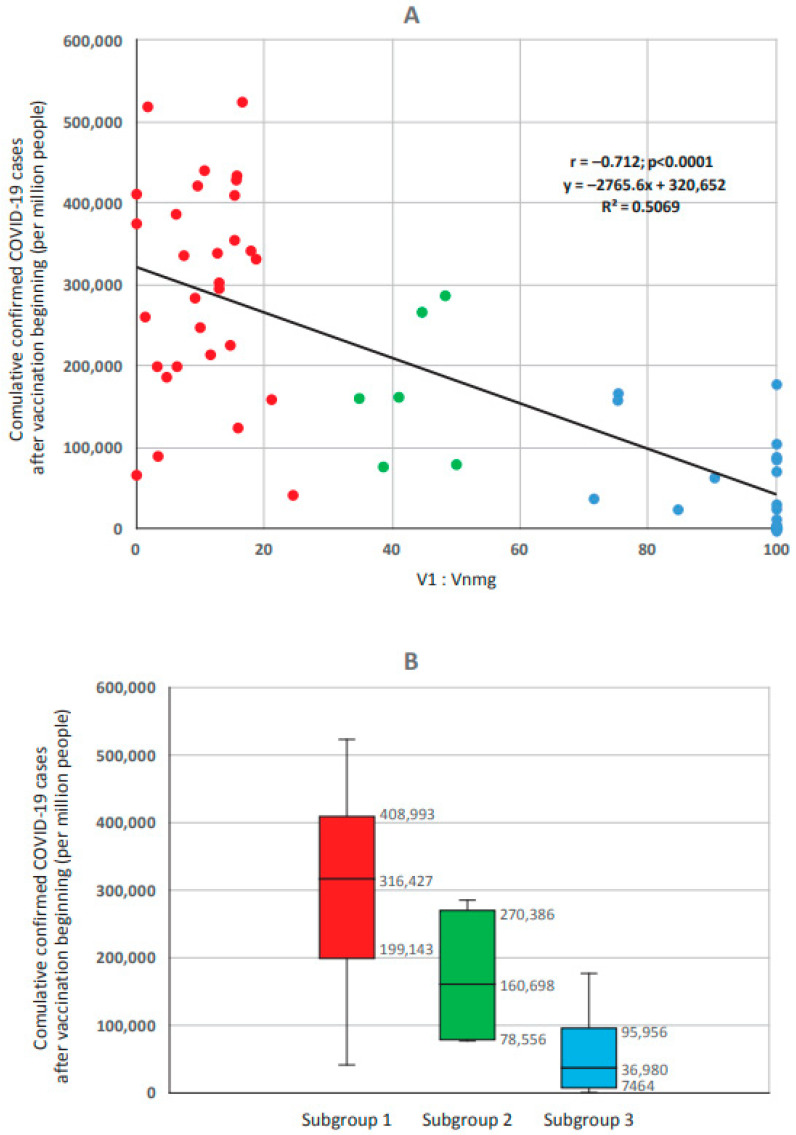
Reported COVID-19 cumulative case numbers after the beginning of vaccination in countries with different V1:Vnmg vaccine usage ratios (*n* = 53). Key: (**A**) shows the correlation between incidence proportion and share Vnmg (r = 0.712; *p* < 0.0001). (**B**) shows the median (Me) and interquartile interval (Q25–Q75) in subgroup 1 (V1 > Vnmg, *n* = 30), subgroup 2 (V1 ≈ Vnmg, *n* = 6), and subgroup 3 (V1 < Vnmg, *n* = 17). Significance (*p*) of differences: *p* (1–2) = 0.036; *p* (2–3) = 0.01; and *p* (1–3) = 0.000001.

**Figure 7 viruses-15-02181-f007:**
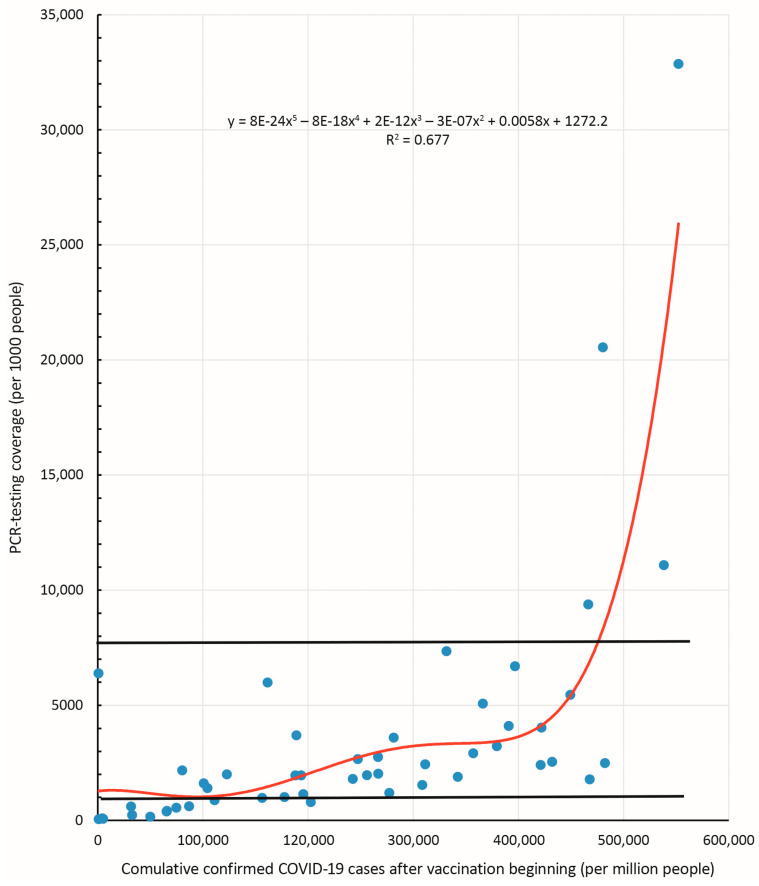
Relationship between SARS-CoV-2 testing coverage (tests per 1000 people) and ‘reported COVID-19 case numbers’ after the beginning of the national vaccination program (per 1 million population). The correlation coefficient and regression equation are shown in the upper left.

**Figure 8 viruses-15-02181-f008:**
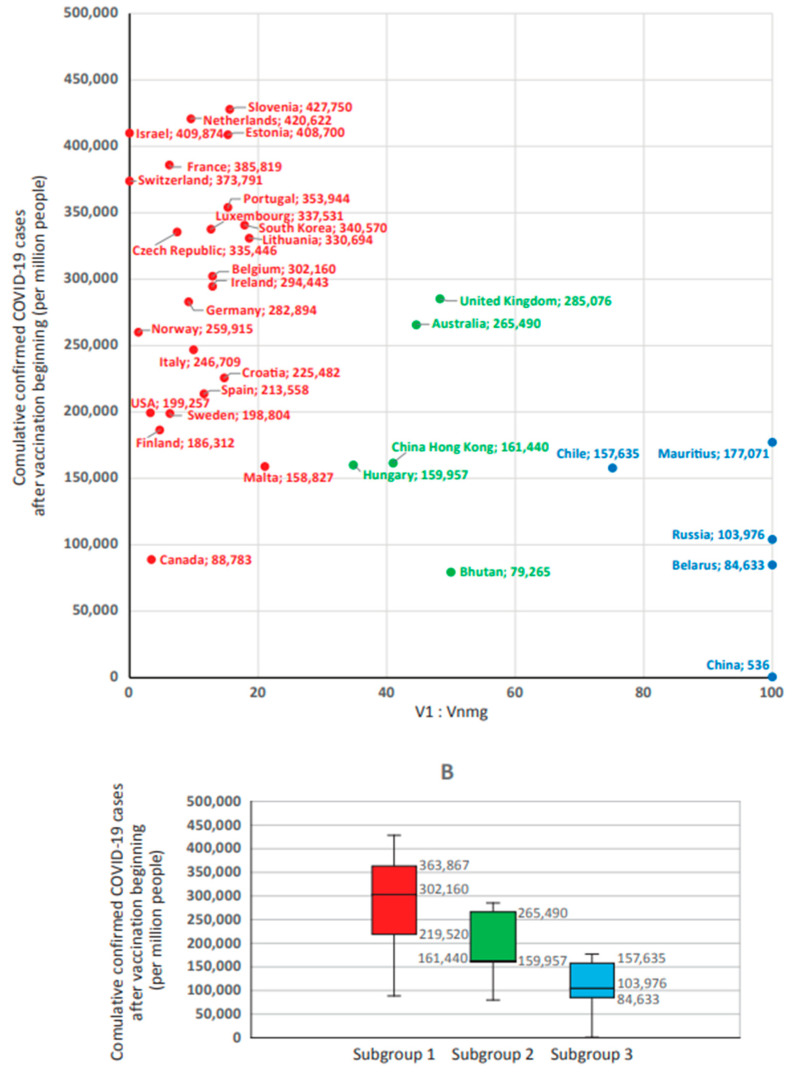
Reported COVID-19 case numbers after the beginning of the national vaccination program in countries with different V1:Vnmg vaccine usage ratios and testing coverage from 1007 to 7348 (per 1000 people) (*n* = 33). Key: (**A**) is distribution of countries by testing coverage. (**B**) is median (Me) and interquartile interval (Q25–Q75) in groups depending on vaccine type dominance: subgroup 1 (V1 > Vnmg, red); subgroup 2 (V1 ≈ Vnmg, green); and subgroup 3 (V1 < Vnmg, blue).

**Figure 9 viruses-15-02181-f009:**
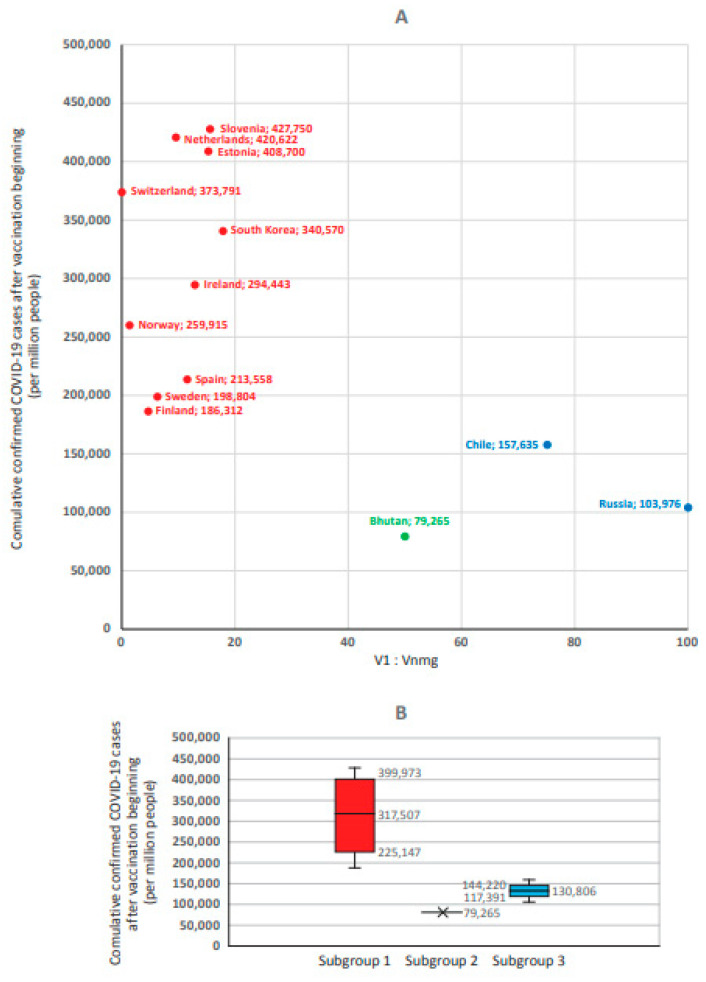
Reported COVID-19 case numbers after the beginning of the national vaccination program in countries (*n* = 13) with different V1:Vnmg vaccine usage ratios and testing coverage near the 2001 level (1697 to 2602 tests per 1000 population). Key: (**A**) shows countries comparable with Russia in terms of testing coverage. (**B**) is median (Me) and interquartile interval (Q25–Q75) in subgroup 1 (V1 > Vnmg, *n* = 11, red); subgroup 2 (V1 ≈ Vnmg, *n* = 1, green); and subgroup 3 (V1 < Vnmg, *n* = 2, blue).

**Figure 10 viruses-15-02181-f010:**
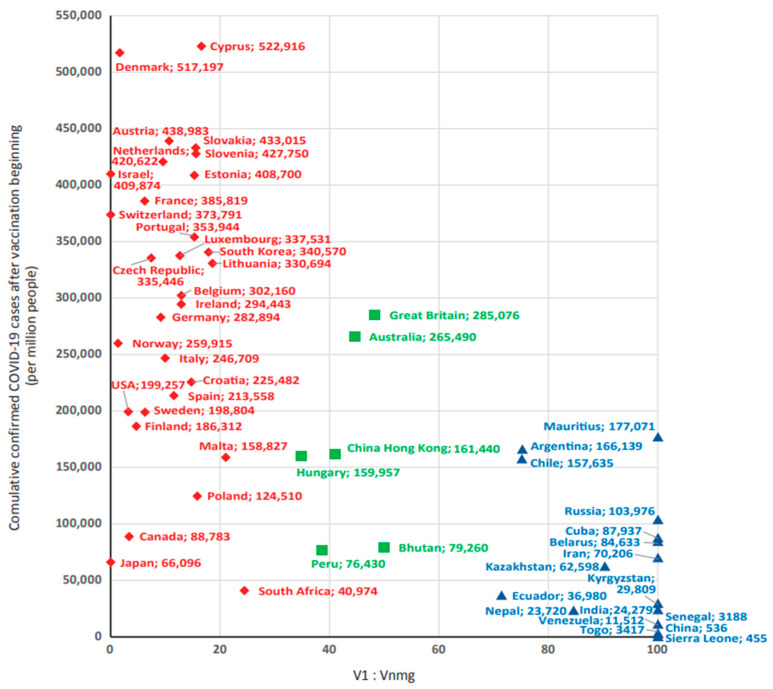
Distribution of countries by ‘number of reported COVID-19 cases’ after the beginning of the national vaccination program and vaccine types used. Note: red diamonds—countries with mRNA vaccine dominance; green squares—countries with near equal use of mRNA and non-mRNA vaccines; blue triangles—countries dominated by non-mRNA vaccines (vector, peptide/protein, whole-virion/inactivated).

**Table 1 viruses-15-02181-t001:** Population size groups and countries (UN classification).

No	Country Group	Population	Number of Countries
1	very large	>100 million	7
2	large	40–100 million	12
3	medium	20–40 million	13
4	small	1.5–20 million	46
5	very small	0.5–1.5 million	8
6	dwarf	<500 thousand	18

**Table 2 viruses-15-02181-t002:** Distribution of countries in the final data set by size group.

No	Country Group	Group Size	Countries
1	very large (>100 million people)	5	India China Russia USA Japan
2	large (40–100 million people)	9	Argentina Great Britain Germany Iran Spain Italy France South Africa South Korea
3	medium (20–40 million people)	6	Australia Venezuela Canada Nepal Peru Poland
4	small (1.5–20 million people)	27	Austria Belarus Belgium Hungary China Hong Kong Denmark Israel Ireland Kazakhstan Kyrgyzstan Cuba Lithuania The Netherlands Norway Portugal Senegal Slovakia Slovenia Sierra Leone Togo Finland Croatia Czechia Chile Switzerland Sweden Ecuador
5	very small (0.5–1.5 million people)	6	Bhutan Cyprus Luxembourg Mauritius Malta Estonia

**Table 3 viruses-15-02181-t003:** Correlation analysis of the adjusted data set (*n* = 53).

Dependent Variable, Significance	Independent Sample Variables							
	Population	Population Density, per km^2^	PCR Testing Coverage, ‰	Vaccination Coverage, %	Vaccine Types Used, %			
					V1	V2	V3	V4
reported COVID-19 case numbers after vaccination beginning	−0.282	−0.017	0.580	−0.1104	0.712	−0.4138	−0.2031	−0.5012
*p*	<0.05	>0.1	<0.001	>0.1	<0.0001	<0.01	>0.05	<0.001

Note. Critical value of the correlation coefficient: 0.233 at *p* = 0.1; 0.276 at *p* = 0.05; 0.358 at *p* = 0.01; and 0.447 at *p* = 0.001.

## Data Availability

All data generated or analyzed during this study are included in this published article and its Appendix A.

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
