# Peer review of "COVID-19 Incidence Proportion as a Function of Regional Testing Strategy, Vaccination Coverage, and Vaccine Type"

_viruses, 2023, doi:10.3390/v15112181_

Round 1

Reviewer 1 Report

Comments and Suggestions for Authors

The review article by Totolian et al describes the analysis regarding the possible relationship between COVID-19 prevalence in the population and the types of SARS-CoV-2 vaccines used in different countries globally, taking into account demographic and epidemiological factors. The study consists of a substantial data collection from 105 countries, followed by a thorough statistical analysis.

Despite the huge efforts taken by the authors in collection and analysis of the data, this study shows some major concerns which need to be addressed before acceptance. 

1.       The majority of documented COVID-19 cases occurred in nations where mRNA vaccinations (V1) predominate, according to the researchers key finding. The nations with the fewest reported cases were those where other vaccines predominated (Vnmg). Further the study found a substantial correlation between testing coverage and COVID-19 prevalence, which makes sense given that more testing will identify more cases of the condition. By carefully analyzing Supplemental Table 2, which lists around 37 nations having testing coverage of 1000 or higher per 1000 population. In almost 90% of these nations with a high number of tests, the mRNA based vaccine predominated. As a result of extensive testing, additional cases have been found in these nations. So, is it reasonable to draw a connection between the number of incidents and the type of vaccination applied in these nations? Please comment.

2. It is not clear about the reported Covid incidences in vaccinated or unvaccinated individual. So, it’s difficult to draw role of different vaccine in prevention of the disease. Also, modulation in severity of disease by the vaccine is an important aspect to determine its protective efficacy.

3. Figure 3 – Figure legends are not legible, also please label the axes.

4. Figure 4 – Figure legends are not legible, also please label the axes.

5. Figure 6, 7 and 8-In these figures also, legends are not legible and axes need to be labelled.

6. Some figures, such as figures 3, 4, 6, 7, and 8, can be provided as supplemental figures.

7. Figure 9 should be placed before figure 10, and the axes should be labelled.

8. The X axis titles of Figure 10 A, 12 A, and 13 A are unclear.

9. Line 325- Corbevax is not a whole virion vaccine, please check and correct.

10. Supplementary tables 1 and 2 should have table headers (column titles) at the top of each page to make reading the tables easier.

Author Response

Answer to Rewiever#1

The majority of documented COVID-19 cases occurred in nations where mRNA vaccinations (V1) predominate, according to the researchers key finding. The nations with the fewest reported cases were those where other vaccines predominated (Vnmg). Further the study found a substantial correlation between testing coverage and COVID-19 prevalence, which makes sense given that more testing will identify more cases of the condition. By carefully analyzing Supplemental Table 2, which lists around 37 nations having testing coverage of 1000 or higher per 1000 population. In almost 90% of these nations with a high number of tests, the mRNA based vaccine predominated. As a result of extensive testing, additional cases have been found in these nations. So, is it reasonable to draw a connection between the number of incidents and the type of vaccination applied in these nations? Please comment.

Answer: Dear reviewer, thank you for bringing this to your attention. Of course, we are aware that the registration of cases depends on the number of tests. We talk about this in the text. To minimize this effect, we conducted a correlation analysis between test coverage and the incidence. As a result, we selected 33 countries with good testing coverage and no correlation (See Fig.11 (form 11)).

  1. It is not clear about the reported Covid incidences in vaccinated or unvaccinated individual. So, it’s difficult to draw role of different vaccine in prevention of the disease. Also, modulation in severity of disease by the vaccine is an important aspect to determine its protective efficacy.

Answer: Of course, you are right. This is an interesting problem. But our study did not set out to study the effect of vaccination on the severity of the disease. This is another study that should be conducted separately.

  1. Figure 3 – Figure legends are not legible, also please label the axes.

Answer: Done

  1. Figure 4 – Figure legends are not legible, also please label the axes.

Answer: Done

  1. Figure 6, 7 and 8-In these figures also, legends are not legible and axes need to be labelled.

Answer: Done

  1. Some figures, such as figures 3, 4, 6, 7, and 8, can be provided as supplemental figures.

Answer: Done

  1. Figure 9 should be placed before figure 10, and the axes should be labelled.

Answer: Done

  1. The X axis titles of Figure 10 A, 12 A, and 13 A are unclear.

Answer: Done

  1. Line 325- Corbevax is not a whole virion vaccine, please check and correct.

Answer: the bug was fixed

  1. Supplementary tables 1 and 2 should have table headers (column titles) at the top of each page to make reading the tables easier.

Answer: Done

Reviewer 2 Report

Comments and Suggestions for Authors

Thanks for submitting this manuscript.

Authors present a “COVID-19 Case Numbers as a Function of Regional Testing Strategy, Vaccination Coverage, and Vaccine Type”, through a very broad database, generated from websites respected worldwide for monitoring the coronavirus.

The authors were able to analyze other variables potentially related to different prevalence rates between different countries, that is, not only vaccination coverage or widespread testing were important for controlling the Pandemic, but they demonstrated data that relate the influence of the types of vaccines that are been used on the prevalence of the COVID-19.

No demerit to any of the types of vaccine in question, but the study contributes to the building of knowledge by the gathering of increasingly greater information or analyses.

The materials and methods, results and discussion were presented properly.

Author Response

Answer to Rewiever#2

Authors present a “COVID-19 Case Numbers as a Function of Regional Testing Strategy, Vaccination Coverage, and Vaccine Type”, through a very broad database, generated from websites respected worldwide for monitoring the coronavirus.

The authors were able to analyze other variables potentially related to different prevalence rates between different countries, that is, not only vaccination coverage or widespread testing were important for controlling the Pandemic, but they demonstrated data that relate the influence of the types of vaccines that are been used on the prevalence of the COVID-19.

No demerit to any of the types of vaccine in question, but the study contributes to the building of knowledge by the gathering of increasingly greater information or analyses.

The materials and methods, results and discussion were presented properly.

Answer: Dear reviewer, thank you very much for the high assessment of our research and for the balanced attitude to the presented research results

Reviewer 3 Report

Comments and Suggestions for Authors

Researchers are presenting us a very interesting study exploring probable relationship between cumulative COVID-19 cases/population and SARS-CoV-2 vaccine types used in different countries of the world, taking into account demographic and epidemiological factors.

There are several methodological issues that have to be taken into account in order to draw conclusion and generalize regarding the interesting findings of the study.

First, rephrase the title and the whole article since you are talking about cases in the title and about prevalence in the manuscript whereas your analysis has to do with cumulative COVID-19 cases/population which is actually incidence proportion.

It would be more important to perform and present/compare the same analysis for deaths due to COVID-19 instead of cases since a) deaths are more stable variable and b) vaccines mainly prevents severe illness and death and secondary prevents from infection.

You have not taken into account different public health measure strategies by each country which is crucial for the number of cases, e.g. lock down periods, contact tracing

You should also mention in your limitation section that reporting strategies differ between countries, not to mention transparency issues.

It is not correct, to my opinion, to count cases from the start of the pandemic that vaccines where not available.

You should also take into account the different waves of the pandemic

Discussion starts with several paragraphs repeating methodological issues, please rephrase.

Comments on the Quality of English Language

Extensive editing of English language is required, esp check axes' labels in tables and figures.

Author Response

Answer to Rewiever#3

Researchers are presenting us a very interesting study exploring probable relationship between cumulative COVID-19 cases/population and SARS-CoV-2 vaccine types used in different countries of the world, taking into account demographic and epidemiological factors.

There are several methodological issues that have to be taken into account in order to draw conclusion and generalize regarding the interesting findings of the study.

First, rephrase the title and the whole article since you are talking about cases in the title and about prevalence in the manuscript whereas your analysis has to do with cumulative COVID-19 cases/population which is actually incidence proportion.

Answer: Done

It would be more important to perform and present/compare the same analysis for deaths due to COVID-19 instead of cases since a) deaths are more stable variable and b) vaccines mainly prevents severe illness and death and secondary prevents from infection.

Answer: We agree that the dependence of mortality on vaccination coverage is an important aspect of evaluating the effectiveness of a vaccine company. However, this requires a separate study that goes far beyond the results presented in the article

You have not taken into account different public health measure strategies by each country which is crucial for the number of cases, e.g. lock down periods, contact tracing

Answer: Of course, we realize that we have not analyzed all the ideally necessary information. Unfortunately, information is sometimes unavailable. We talk about this in the section «Limitations of the study»

You should also mention in your limitation section that reporting strategies differ between countries, not to mention transparency issues.

Answer: Done

It is not correct, to my opinion, to count cases from the start of the pandemic that vaccines where not available.

You should also take into account the different waves of the pandemic

Answer: we mentioned these issues in the limitation section

Discussion starts with several paragraphs repeating methodological issues, please rephrase.

Answer: We have repeated the main methodological issues to make it easier to follow the discussion. However, if you insist, we could delete them.

Round 2

Reviewer 3 Report

Comments and Suggestions for Authors

Dear Authors,

Thank you for taking into consideration several of my suggestions.

Please rephrase prevalence with incidence proportion in Figures 1 and 2.

Answer: We agree that the dependence of mortality on vaccination coverage is an important aspect of evaluating the effectiveness of a vaccine company. However, this requires a separate study that goes far beyond the results presented in the article.

Please refer to this in the limitation section.

It is not correct, to my opinion, to count cases from the start of the pandemic that vaccines where not available.

Answer: we mentioned these issues in the limitation section

I think it is not a matter of limitation. You can use the correct number of COVID-10 cases for the period after onset of vaccination. Otherwise your conclusions are not valid.

Comments on the Quality of English Language

Extensive editing of English language required

Author Response

Thank you for taking into consideration several of my suggestions.

Please rephrase prevalence with incidence proportion in Figures 1 and 2.

Answer: Dear Reviewer,

Thank you again for your comments and suggestion. Thanks to them, the article has become much better. We have adjusted all the figures in accordance with your recommendations

Answer: We agree that the dependence of mortality on vaccination coverage is an important aspect of evaluating the effectiveness of a vaccine company. However, this requires a separate study that goes far beyond the results presented in the article.

Please refer to this in the limitation section.

Answer: Done

It is not correct, to my opinion, to count cases from the start of the pandemic that vaccines where not available.

Answer: we mentioned these issues in the limitation section

I think it is not a matter of limitation. You can use the correct number of COVID-10 cases for the period after onset of vaccination. Otherwise your conclusions are not valid.

Answer: We have recalculated all values since the beginning of national vaccination programs. The corresponding changes are marked in green. We also adjusted all the figures to take into account the recalculated values